# Diversity and Biotechnological Potential of Cultivable Halophilic and Halotolerant Bacteria from the “Los Negritos” Geothermal Area

**DOI:** 10.3390/microorganisms12030482

**Published:** 2024-02-27

**Authors:** Joseph Guevara-Luna, Ivan Arroyo-Herrera, Erika Yanet Tapia-García, Paulina Estrada-de los Santos, Alma Juliet Ortega-Nava, María Soledad Vásquez-Murrieta

**Affiliations:** 1Escuela Nacional de Ciencias Biológicas, Instituto Politécnico Nacional, Mexico City 11340, Mexico; josephguevara.28@gmail.com (J.G.-L.); jw.17@hotmail.com (I.A.-H.); erikatenay@gmail.com (E.Y.T.-G.); pestradadelossantos@gmail.com (P.E.-d.l.S.); 2Centro de Estudios de Bachillerato “José Vasconcelos”, Iguala 40000, Mexico; almajortegan@hotmail.com

**Keywords:** bioprospection, halophilic bacteria, halotolerant bacteria, geothermal zone

## Abstract

Soil salinization is negatively affecting soils globally, and the spread of this problem is of great concern due to the loss of functions and benefits offered by the soil resource. In the present study, we explored the diversity of halophilic and halotolerant microorganisms in the arable fraction of a sodic–saline soil without agricultural practices and two soils with agricultural practices (one sodic and one saline) near the geothermal area “Los Negritos” in Villamar, Michoacán state. This was achieved through their isolation and molecular identification, as well as the characterization of their potential for the production of metabolites and enzymes of biotechnological interest under saline conditions. Using culture-dependent techniques, 62 halotolerant and moderately halophilic strains belonging to the genera *Bacillus*, *Brachybacterium*, *Gracilibacillus*, *Halobacillus*, *Halomonas*, *Kocuria*, *Marinococcus*, *Nesterenkonia*, *Oceanobacillus*, *Planococcus*, *Priestia*, *Salibactetium*, *Salimicrobium*, *Salinicoccus*, *Staphylococcus*, *Terribacillus*, and *Virgibacillus* were isolated. The different strains synthesized hydrolytic enzymes under 15% (*w*/*v*) of salts, as well as metabolites with plant-growth-promoting (PGP) characteristics, such as indole acetic acid (IAA), under saline conditions. Furthermore, the production of biopolymers was detected among the strains; members of *Bacillus*, *Halomonas*, *Staphylococcus*, and *Salinicoccus* showed extracellular polymeric substance (EPS) production, and the strain *Halomonas* sp. LNSP3E3-1.2 produced polyhydroxybutyrate (PHB) under 10% (*w*/*v*) of total salts.

## 1. Introduction

Biodiversity in saline environments is limited due to high concentrations of salt. Saline soils have their own microbial communities, designated as halophiles, that have adapted to high salt content. Halophiles are extremophilic microorganisms that cope with several environmental factors, such as alkaline pH, low oxygen availability, fluctuating temperatures, and heavy metals, besides salinity [1].

Halophiles can be divided into (i) extreme (2.5–5.9 M~15–32% NaCl), (ii) moderate (0.5–2.5 M~3–15% NaCl), and (iii) slight (0.2–0.5 M~1–3% NaCl); however, there are halotolerant microorganisms, which do not require NaCl for growth but can tolerate even high concentrations of this and other salts [1,2]. The survival of these microorganisms in hypersaline conditions requires specialized cellular and enzymatic adaptation to preserve the osmotic balance with the environment [3].

The role of halophilic and halotolerant microorganisms in the environments where they are found is highly dynamic and important for biogeochemical cycles in conditions of high salinity [4]. The physiological characteristics of halophiles contribute to their usefulness, and they are seen as a source of biomolecules, biomaterials, and metabolites. In recent years, the biotechnological applications of halophilic microorganisms have increased in number, and others are under development, such as in (i) the production of extreme enzymes, (ii) the production of ectoine, (iii) the production of exopolysaccharides, (iv) the production of bioplastics (polyhydroxyalkanoates [PHAs]), and (v) plant-growth-promoting (PGP) activity [5,6].

Soil salinization is predicted to affect more than 50% of the total cultivable soil worldwide by the year 2050 [7]. Due to their survival in high and fluctuating salt concentrations and their versatility in producing biomolecules or natural products of biotechnological interest, halophilic and halotolerant microorganisms in extreme natural environments show promise for the development of sustainable strategies for the recovery of saline environments or the improvement of industrial processes [8].

“Los Negritos” is a geothermal area that has been studied in terms of soil and water characterization to evaluate salt content, as well as geothermal activity as a potential alternative energy source [9,10]. There are few reports on the diversity of microorganisms inhabiting the soils of the geothermal area. Guevara-Luna et al. [11] investigated the archaea and bacteria community diversities through high-throughput sequencing analysis of the 16S rRNA gene and the physicochemical characteristics that could explain their relative abundance in these soils. On the other hand, Pérez-Inocencio et al. [12] isolated the bacteria associated with the rhizosphere of halophytes in this area, which showed different PGP activities and synthesis of hydrolytic enzymes.

Therefore, this study aimed to investigate the cultivable halophilic and halotolerant bacterial diversity in a saline–sodic soil without agricultural use and two soils destined for agricultural practices (one sodic and one saline) from “Los Negritos”, Villamar, Michoacán, and to investigate the microorganisms’ potential to produce indole acetic acid (IAA), siderophores, exopolysaccharides, PHAs, and enzymes of biotechnological interest under saline conditions.

## 2. Materials and Methods

### 2.1. Soil Sampling

Samples from two arable soils (Arable soil 1 [AS1] and Arable soil 2 [AS2]) and a natural saline–sodic soil (Saline [S]) without agricultural practices were collected from the geothermal zone “Los Negritos”, Villamar, Michoacán state, in March 2018 as described by [11]. The geographic locations of AS1, S, and AS2 were 20°02.810′ N 102°36.996′ W, 20°03.780′ N 102°36.819′ W, and 20°03.655′ N 102°36.655′ W, respectively. Briefly, each soil was divided into three plots, and samples were collected from nine points in each and mixed to obtain one composite sample from each plot (500 g, *n* = 9). These were transferred to polyethylene bags and transported to the laboratory for further analysis [11].

### 2.2. Bacterial Enrichment and Isolation

To isolate halophilic and/or halotolerant microorganisms from the soil samples, two strategies were applied. The first method, an enrichment protocol [13], was performed using SP or halophilic medium (HM) broth [14] supplemented with 10% and 22% NaCl (*w*/*v*), respectively. The enrichment of halophilic and halotolerant bacteria was performed by transferring 1 g of soil from each composite soil sample to flasks containing 20 mL of SP or HM broth and incubating them at room temperature (28 °C ± 2 °C) for 7 days at 150 rpm. This process was repeated three times (i.e., 1 mL of broth was transferred to a flask with fresh broth at the end of the first and second week of enrichment), and the final enrichment culture was used for isolation using an aliquot of 100 µL dispersed in SP or HM agar plates and incubated at 30 °C for 7 days. Colonies with different morphologies were selected and purified by repeated streaking on SP or HM plates.

The second method used was direct bacterial isolation from the composite soil samples (without enrichment) by serially diluting 1 g of soil in sterile NaCl 10% (*w*/*v*) solution until 10^−^^5^. The last three dilutions (10^−^^3^ to 10^−^^5^) were plated onto SP 10% NaCl, HM 22% NaCl, and trypticase soy agar (TSA; DIFCO, Franklin Lakes, NJ, USA) plates twice and incubated for 7 days at 28 °C. The total colonies were counted to estimate the CFUs per gram of soil, and colonies with different morphologies were selected and transferred to SP or HM agar plates until axenic cultures were obtained. The purified bacterial cells from both isolation techniques were preserved with (i) 50% glycerol and (ii) a mixture of 80% glycerol, 6% saline water, and 1 mM CaCl_2_ solution at −72 °C [15,16].

### 2.3. Microscopy and NaCl Tolerance Characterization

A modified Gram stain was performed as described by [17] for the microscopical characterization of the axenic isolates. To evaluate the tolerance of the isolates for NaCl, SP or HM plates with different concentrations of NaCl (0% up to 27.5% [*w*/*v*] at intervals of 2.5%) were prepared [18,19].

### 2.4. Genomic DNA Extraction and Phylogenetic Analysis

Genomic DNA (gDNA) extraction was performed according to a modified protocol described previously [20], and the DNA was used as a template for the amplification of the 16S rRNA gene. The 25 µL PCR reaction mixture contained 1 µL of gDNA, 0.5 µL of universal primers (27 F and 1492 R at 10 pmol) [21], 0.5 µL of dNTPs (10 mmol), 2.5 µL of Thermo Scientific™ 10× DreamTaq™ Buffer (20 mmol MgCl_2_), 3 µL of 25 mmol MgCl_2_, 2 µL of BSA (10 mg mL^−1^), and 0.125 µL of DreamTaq DNA polymerase (Thermo Fisher Scientific, Waltham, MA, USA). The thermocycling conditions included an initial denaturation step at 94 °C for 5 min; followed by 30 cycles at 94 °C (1 min), 56 °C (30 s), and 72 °C (2 min); and one final cycle at 72 °C for 10 min. The amplicon quality was verified by gel electrophoresis in 1% agarose. Sequencing was performed by Macrogen Inc. (Seoul, Republic of Korea).

To determine the clonality of the isolates, molecular typing using BOX-PCR was performed. The 20 µL reaction mixture consisted of 1 µL of gDNA, 0.4 µL of dNTPs (10 mmol), 1.2 µL of 25 mmol MgCl_2_, 2 µL of Thermo Scientific™ 10× DreamTaq™ Buffer (20 mmol MgCl_2_), 1 µL of BOXA1R primer (5′-CTACGGCAAGGCGACGCTGACG-3′) [22,23], 0.8 µL of BSA (10 mg mL^−1^), and 0.125 µL of DreamTaq DNA polymerase (Thermo Fisher Scientific). The thermocycling conditions comprised denaturation at 95 °C for 5 min; 35 cycles at 95 °C (1 min), 63 °C (1 min), and 72 °C (6 min); and a final extension at 72 °C for 10 min [24]. The amplicon patterns were visualized with gel electrophoresis in 1.5% agarose.

The partial 16S rRNA gene sequences were edited manually using the Seaview4 software [25]. Sequences with a threshold of ≥97% were downloaded from the EzBioCloud database (version 2023.08.23) [26] and used for multiple sequence alignment with the UGENE software v. 1.30.0 using the MUSCLE algorithm [27]. The aligned sequences were used for the estimation of the evolutionary model with the jModelTest 2 software [28], and a similarity matrix was constructed with the MatGAT 2.02 software [29].

A phylogenetic tree was constructed using the Bayesian inference method with the Beast software v. 2.6 [30] with the GTR + I + G model. A total of 10^7^ generations were performed, after which 25% of the trees were discarded [31]. *Acidobacterium capsulatum* ATCC 51196 (CP001472) and *Acidobacterium ailaaui* PMMR2 (JIAL01000001) were used as outgroups.

The 16S rRNA sequences of the isolated strains were submitted to the GenBank database (https://www.ncbi.nlm.nih.gov/genbank/; submission date 14 September 2023) and provided with accession numbers OR553742 to OR553803.

### 2.5. Hydrolytic Activity

The activity of different hydrolytic enzymes (amylases, cellulases, inulinases, lipases, esterases, DNAses, pectinases, and proteases) under 15% NaCl (*w*/*v*) was tested in agar plates supplemented with appropriate substrates according to [32,33] as well as by adding the substrates to SP or HM plates. Each plate was inoculated with 10 µL of a 48–72 h culture of SP or HM broth incubated as previously described. The hydrolytic activity was expressed semi-quantitatively as levels of enzymatic activity (LEA) according to [32,34].

### 2.6. Plant-Growth-Promoting Characteristics

Screening for IAA production was performed qualitatively using a modified Jain and Patriquin (JP) medium [35] supplemented with 10% NaCl according to [36]. To quantify IAA production, isolates with positive activity were inoculated in SP or HM broth, and the cells were adjusted to an O.D. of approximately 0.15–0.2 absorbance, and 100 µL of suspension was inoculated in JP or HM broth (supplemented with 0.1 g L^−1^ of L-tryptophan) with 10% NaCl and in the same broths without NaCl. The isolates were incubated for 7 days at 28 °C and 150 rpm. The cell-free supernatants were obtained at 4500 rpm, and an aliquot of 1 mL was mixed with 1 mL of the Salkowski solution [37]. The IAA standard (Sigma-Aldrich, St. Louis, MO, USA) was used to construct a calibration curve from 0 to 1000 µg mL^−1^ spectrophotometry at 540 nm using a Hach DR 2800 (HACH, Loveland, CO, USA). *Azospirillum brasilense* Sp7 was used as positive control without NaCl addition.

Siderophore production was tested qualitatively using SP or HM agar plates supplemented with 10% NaCl and chrome azurol S solution [38]. The plates were inoculated with 5 µL of culture broth grown using the conditions previously described and incubated at 30 °C for 7 days. Colonies with an orange ring around them were positive for siderophore production [39,40]. All the screening determinations were performed in triplicate.

### 2.7. Extracellular Polymeric Substance Production

The isolates were inoculated in SP 10% NaCl or HM 22% NaCl broth as previously described. One hundred microliters was streaked on ATCC N° 14 and malt yeast (MY) agar plates supplemented with 10% NaCl for 6 days at 30 °C. The presence of mucoid colonies on the agar surface was considered a positive result for extracellular polymeric substance (EPS) production [41].

### 2.8. Polyhydroxyalkanoate Production

To evaluate PHA synthesis and accumulation by the halophilic or halotolerant bacteria, NaCl synthetic medium (NSM) [42] and modified accumulation medium (MAM) [43] supplemented with 10% NaCl and 0.01% Nile red were used to inoculate by streaking 10 µL on agar plates and incubating for 14 days at 30 °C. PHA accumulation was observed by exposing the plates to UV light (302 nm); those with pink-orange fluorescence were considered positive for production [42].

### 2.9. Polyhydroxyalkanoate Extraction and Partial Characterization

The promising PHA-synthesizing strain was inoculated on MAM (10% NaCl) agar plates and incubated at 30 °C until growth was observed. From this plate, the strain was inoculated in 10 mL of SP 10% NaCl broth for 24 h at room temperature (150 rpm). The cells were washed three times with a 10% NaCl sterile solution to eliminate SP broth residue. Then, 2 mL of the suspension adjusted to an absorbance value of 1 (O.D.) was inoculated in 200 mL of MAM 10% NaCl medium with 2% glucose in triplicate. The flasks were incubated at 26–28 °C and 150 rpm for 120 h. The polymer was extracted according to [44] and quantified as described by [45]. The sample was concentrated and analyzed by Fourier-transform infrared spectroscopy (FT-IR). A polyhydroxybutyrate (PHB) standard (Sigma-Aldrich) was used as a control.

### 2.10. Statistical Analysis

Analysis of variance (ANOVA) with the Tukey’s honest significant difference (HSD) test was performed using R software v. 4.0.5 and RStudios v. 1.4.1106 [46,47] with the aov and HSD.test functions, respectively. A principal component analysis (PCA) was conducted to visualize the correlation between phenotypic features and physicochemical soil characteristics using the FactoMineR package [48]. A heatmap of the different hydrolytic enzyme activities was created using the pheatmap package [49]. The *t*-test for two samples was used to compare the culture conditions for IAA production at a significance value of *p* < 0.05.

## 3. Results

### 3.1. Halophilic and Halotolerant Microorganisms from “Los Negritos”

The number of viable microorganisms per gram of soil was 4.4 × 10^7^ CFU on TSA plates and 6 × 10^5^ CFU on SP plates, estimating the number of halotolerant microorganisms (Appendix A). Furthermore, using both enrichment and direct bacterial isolation through serial dilution, a total of 62 bacterial isolates, 51 in SP 10% medium (82%) and 11 in HM 22% NaCl medium (18%), were isolated. Out of the 62 isolates, 33 were Gram-positive rods, 11 were Gram-negative rods, and 18 were Gram-positive coccoid. A total of 74.2% (46/62) of the isolates were halotolerant, while 25.8% (16/62) showed a requirement of at least 2.5% NaCl for growth (Appendix A).

Three phyla were identified through 16S rRNA sequencing: Bacillota (69.4%, 46/62) was the dominant phylum, followed by Pseudomonadota (17.7%, 11/62) and Actinomycetota (8.1%, 5/62; Figure 1), with the similarity between the sequences ranging from 90% to 99% (Appendix A). Among the Bacillota members, Oceanobacillus was predominant, followed by Bacillus, Gracilibacillus, Halobacillus, Marinococcus, Planococcus, Priestia, Salibactetium, Salimicrobium, Salinicoccus, Staphylococcus, Terribacillus, and Virgibacillus. Members of Halomonas were dominant among the Pseudomonadota, and the Brachybacterium, Kocuria, and Nesterenkonia genera were found among Actinomycetota (Appendix A).

The BOX-PCR analysis showed that members of the dominant genus (i.e., Oceanobacillus) showed eight different band patterns between the 14 isolated strains. Members of Halomonas showed six different band patterns between the 11 strains, whereas Salimicrobium and Staphylococcus showed two and three different band patterns, respectively (Appendix A).

The “Los Negritos” soils showed some interesting characteristics; the cation exchange capacity (CEC) and total nitrogen (N) content of AS1 were 46.8 cmol kg^−1^ and 1.8 g kg^−1^ dry soil, respectively. The saline soil had a pH value of 9.2 and electrolytic conductivity (EC) of 17.6 dS m^−1^, while the AS2 soil showed a high content of arsenic (As; 175 mg kg^−1^ dry soil), cadmium (Cd; 5.7 mg kg^−1^ dry soil), and nitrates (NO_3_^−^; 39.7 mg kg^−1^ dry soil) [11]. Considering the above values, in this work the PCA analysis showed that the analyzed physicochemical characteristics of the “Los Negritos” soil explained 100% of the data variation and the association between these and the isolated strains (PC1 61.5% and PC2 38.5%; Figure 2). Members of *Oceanobacillus*, *Terribacillus*, and *Virgibacillus* were enriched in AS1, while members of *Halomonas*, *Salibacterium*, and *Salimicrobium* were enriched in the saline soil, and members of *Brachybacterium*, *Nesterenkonia*, and *Staphylococcus* were enriched in AS2 (Figure 2).

### 3.2. Biotechnological Potential of the Halotolerant and Halophilic Bacteria

A total of 60 out of 62 strains synthesized hydrolytic enzymes under 15% NaCl (*w*/*v*). Inulinase and esterase activities (Tween 20 and 80) were the most frequent among the strains, followed by those of lipases, DNAses, and proteases (Figure 3). Only four strains showed activity of two or more hydrolytic enzymes with a medium LAE value between 2 and 5, that is, *Marinococcus* sp. LNHM5E3-2.1, *Nesterenkonia* sp. LNSP9103-1, and *Salimicrobium* sp. LNHM3E3-1.1 and LNHM10E3-1. The largest number of enzyme-producing strains were isolated from AS1, followed by S and AS2 (Figure 3).

The qualitative and quantitative screening of the IAA producer strains under 10% NaCl (*w*/*v*) showed that 17.7% (11/62) could produce the auxin. *Salibacterium* sp. LNHM5E3-1 and LNHM5E3-2.2 had values of 7.9 and 9.18 µg mL^−1^, respectively. *Salibacterium* sp. strain LNHM5E3-2.2 was selected for IAA production under 10% NaCl in JP and HM supplemented with L-tryptophan (HMt) and without salinity conditions. The LNHM5E3-2.2 strain showed a statistically significant difference in IAA production between the JP and HMt broths under 10% NaCl (*p* = 0.045) with a maximum at 120 h, indicating higher IAA production in the JP broth (Appendix A) following further experiments in this broth. Furthermore, statistical analysis showed no significant difference in the growth of the LNHM5E3-2.2 and *A. brasilense* Sp7 strains without NaCl but revealed a difference in IAA production (*A. brasilense* 29.7 ± 0.04 μg mL^−1^, *Salibacterium* sp. LNHM5E3-2.2 26.8 ± 3.2 μg mL^−1^; *p* = 0.022). In the presence of 10% NaCl, *Salibacterium* sp. LNHM5E3-2.2 showed 10.54 ± 1.5 μg mL^−1^ of IAA at 168 h, which was less than the amount in experiments without NaCl (Appendix A).

Siderophores were produced by 38.7% (24/62) of the isolated strains under 1% NaCl, whereas only 6.4% (4/62) of the strains could produce siderophores under 10% NaCl (Appendix A).

Of the 62 strains, 9 showed orange fluorescence in the MAM plates with 10% NaCl, of which 8 were members of the *Halomonas* genus. To the best of our knowledge, there are no reports of members of *Halomonas* from “Los Negritos” being PHA producers; therefore, we studied PHA production by *Halomonas* sp. LNSP3E3-1.2 in depth. The FT-IR analysis showed characteristic peaks for PHB polymer produced by the LNSP3E3-1.2 strain compared with the PHB standard. We observed stretching at 3289.7 cm^−1^ corresponding to -OH functional groups, 2974.7 and 2933.5 cm^−1^ corresponding to -CH groups, and 1724.3 cm^−1^ corresponding to an ester group (C=O), also observed in the PHB standard (1725.1 cm^−1^). The peaks at 1380.9 cm^−1^, 1281.4 cm^−1^, 1132.3 cm^−1^, and 1056.9–979.6 cm^−1^ corresponded to -CH3, -CH2, C-O-C, and -CH groups, respectively (Figure 4).

In total, 11.3% (7/62) of the strains were positive for EPS production in the two media used. Four out of seven strains showed EPS production on the ATCC 14 medium with 10% NaCl (i.e., *Bacillus* sp. LNSP2103-3 and LNSP2103-4, *Oceanobacillus* sp. LNSP3E3-1 and LNSP3E3-2), whereas three out of seven showed a positive result on the MY medium with 10% NaCl (*Bacillus* sp. LNSP2103-3 and LNSP2103-4, *Salinicoccus* sp. LNSP6E3-1).

## 4. Discussion

The soils of the “Los Negritos” geothermal area are used for agricultural practices; however, due to the irrigation of the fields with water containing salts (>2 dS m^−1^), soil salinization has occurred (>40 dS m^−1^) [10,50].

In this study, the number of viable microorganisms observed showed that these can cope with saline stress. Soil microorganisms constitute <0.5% (*w/w*) of the soil mass and carry out key functions in the nutrient cycle [51]. The two culture-dependent strategies used in this study (enrichment and serial dilutions) using two media with different compositions and salt concentrations allowed the isolation of a total of 62 strains. Differences in the number of strains isolated from SP and HM media could be due to differences in their composition. The SP medium has a medium salt concentration (10% *w*/*v*), high N content, and an easily assimilated carbon source (glucose), while the HM medium has a higher salt content (22% *w*/*v*) and is suggested primarily for the isolation of halophilic archaea [52].

The similarity analysis of the 16S rRNA sequences showed a total of 17 genera among the isolated strains, with a similarity of ≤98% compared with the type strain of each identified genus. According to [53], the similarity value between two sequences proposed to identify an isolate as a species is 98.7–99%. The phylogenetic tree reconstruction showed that the isolated strains in this study differed from the reported species, suggesting that some of these strains could be new species; however, polyphasic analyses are necessary for confirmation.

Bacillota was the dominant phylum, followed by Pseudomonadota and Actinomycetota. This could be explained by the ability of members of Bacillota (e.g., *Bacillus*) to synthesize desiccation-resistant proteins and form endospores [54], as well as the synthesis or internalization of compatible solutes (e.g., betaine, ectoine, hydroxyectoine, and polyalcohols) by members of Pseudomonadota and Actinomycetota [55]. Interestingly, Guevara-Luna et al. [11] found that *Serratia* and *Bacillus* were the most abundant genera in the soils of “Los Negritos”, along with *Halomonas*, through high-throughput sequencing analysis of the 16S rRNA gene, similar to the results of this study.

The PCA showed a relationship between the members of *Oceanobacillus* and the Fe and Cu content. Maity et al. [56] observed that the *Oceanobacillus indicireducens* 5(225) strain resisted up to 600 mg L^−1^ of Cu, which could be explained by oxidoreductase-mediated mechanisms, as reported by [57]. The isolated strains of *Terribacillus* showed a correlation with the total N and Pb content. Orhan and Demirci [58] reported that *Terribacillus saccharophilus* (EM15) presented 1-aminocyclopropane-1-carboxylic acid (ACC) deaminase activity, which could explain the correlation with the total N content, suggesting that members of *Terribacillus* from “Los Negritos” may have this enzymatic activity.

Delgado-García et al. [59] found a correlation between the Na, K, Mg, CO_3_^−^, HCO_3_^−^, and Cl^−^ content and the genera *Alkalibacillus*, *Bacillus*, *Halobacillus*, and *Marinococcus*, as observed for *Bacillus* and *Halobacillus* in this study. Arayes et al. [60] found that members of *Bacillus* were more abundant in moderate salinity and the presence of SO_4_^2−^, whereas members of *Salinicoccus* required K and Mg, correlations similar to those observed in this study. The strains correlated with factors associated with salinity (e.g., pH, EC, CEC, and ion content) belonged to halotolerant genera that possessed osmoadaptative mechanisms [61,62] for coping with up to 20% NaCl. These findings concurred with the results of this study, since most of the strains isolated from the saline soils of “Los Negritos”, Michoacán tolerated salt concentrations even above 20% (*w*/*v*) NaCl.

Several bacteria and archaea isolated from saline soils in different parts of the world can produce stable enzymes at high NaCl concentrations, most of which were detected in this study [32,33]. High-molecular-weight biopolymer hydrolysis is a key step in the metabolism of organic compounds in the ecosystem, playing an important role in the biogeochemical cycle of nutrients in saline soils; thus, the study of the enzymatic activities of microorganisms present in areas affected by salinity is of great interest due to the great biotechnological potential of these enzymes [63]. Studies have reported the ability of halophilic and halotolerant bacteria isolated from saline environments to synthesize a wide variety of hydrolytic enzymes of industrial interest, as seen in this study [32,33,64].

A total of nine strains of *Halomonas* showed the ability to produce PHAs under saline conditions, which is in accordance with other reports on members of *Halomonas*, which are considered important PHA producers and accumulators [65]. Therefore, it was interesting to confirm and characterize the production of this polymer by the *Halomonas* sp. LNSP3E3-1.2 strain. The FT-IR analysis confirmed the production of PHB by strain LNSP3E3-1.2; other studies have also reported members of genus *Halomonas* as producers of PHAs with spectra similar to those observed in this study [66,67].

In this study, members of Bacillota produced EPS under high salt content. The bacterial synthesis of these polymers under osmotic stress (e.g., high concentrations of salts) is associated with a tolerance mechanism [68].

The halotolerant and moderately halophilic strains in this study derived from agricultural and saline soils produced IAA and siderophores under saline conditions (10% *w*/*v* NaCl). Goswami et al. [69] evaluated the PGP characteristics of 85 rhizospheric isolates, of which *Bacillus licheniformis* A2 showed the best characteristics, including IAA production at a concentration of 1 M NaCl (~6%) in a medium supplemented with L-tryptophan; the growth of *Arachis hypogaea* was also enhanced at 50 mM NaCl. Nghia et al. [70] reported the strain *Bacillus megaterium* ST2-9 (now reclassified as *Priestia megaterium*), with the potential to promote plant growth under 3% NaCl. Throughout the world, the problem of salinity in arable soils in arid and semi-arid regions is growing, negatively and directly affecting plant growth [71]. The PGP characteristics of the strains isolated in this work could help in the development of strategies to promote agriculture under the extreme conditions of saline soils as reported by [12]. Furthermore, the strains isolated in this work are representatives of well-known microbial groups; however, the findings in this study suggest that these strains could be new representatives of these microbial groups, and more detailed studies are required to confirm that observation. On the other hand, the findings in this study indicate that the strains or their metabolites could be applied to agriculture in soils with high salt concentrations to improve the development of crops. The hydrolytic enzymes are promising for industrial applications such as the development of biodegradable detergents or treatment of residual water, but more studies are needed.

## 5. Conclusions

The diversity of cultivated halophilic and halotolerant bacteria from “Los Negritos” soils was greater in the AS2 site compared to the agricultural (AS1 and AS2) sites. The isolated strains comprised genera that included possible new species, mainly related to salinity and other physicochemical factors, such as metal and total nitrogen content, which may favor their presence. On the other hand, the strains reported in this study had the ability to synthesize metabolites such as hydrolytic enzymes, EPS, and PHAs and had PGP characteristics (IAA, siderophores) under saline conditions, which are of great biotechnological interest with a wide range of applications.

## Figures and Tables

**Figure 1 microorganisms-12-00482-f001:**
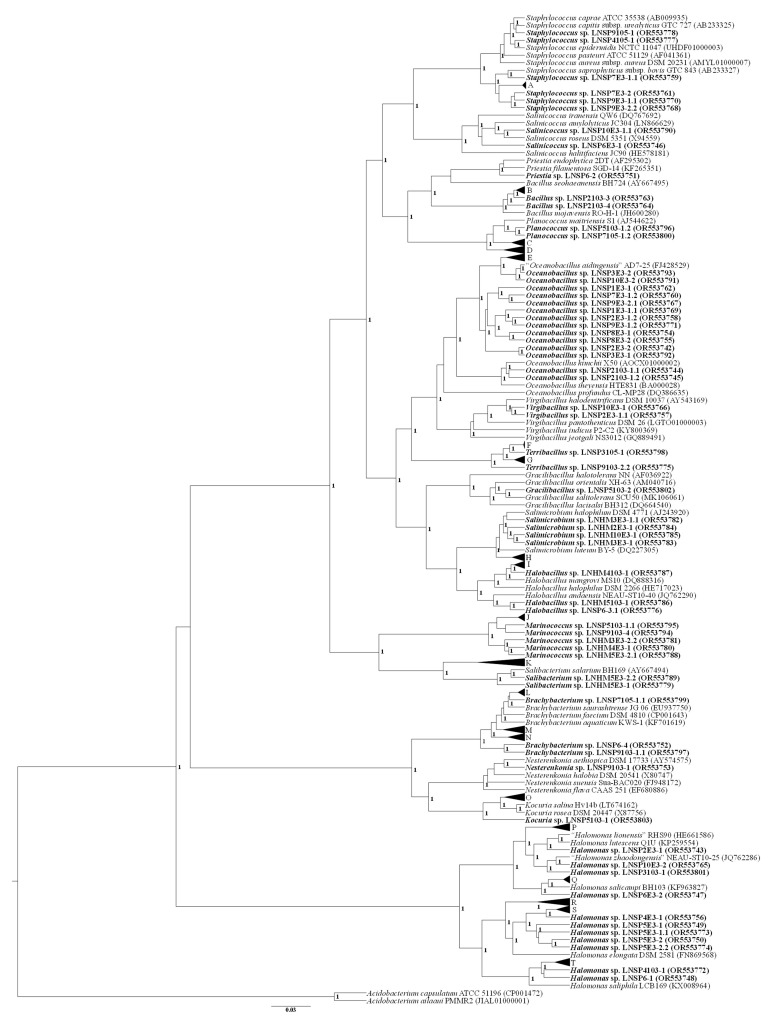
Phylogenetic tree of the 16S rRNA sequences using the Bayesian inference method with the GTR + I + G model. The bold names show the strains isolated in this study. Uppercase letters indicate collapsed clades as follows: (A) *Staphylococcus saprophyticus* subsp. *saprophyticus* ATCC 15305 (AP008934), *Staphylococcus pseudoxylosus* S04009 (MH643903), *Staphylococcus edaphicus* P5085 (KY315825); (B) *Bacillus velezensis* CR-502 (AY603658), *Bacillus halotolerans* ATCC 25096 (LPVF01000003); (C) *Planococcus rifietoensis* M8 (CP013659), *Planococcus citreus* DSM 20549 (RCCP01000013); (D) *Planomicrobium soli* XN13 (JQ772482), *Planococcus ruber* CW1 (KX950835); (E) *Oceanobacillus oncorhynchi* subsp. *oncorhynchi* R-2 (AB188089), “*Oceanobacillus jeddahense*” S5 (CCDM010000002), *Oceanobacillus oncorhynchi* subsp. *incaldanensis* 20AG (AJ640134); (F) *Terribacillus saccharophilus* 002-048 (AB243845), *Terribacillus goriensis* CL-GR16 (DQ519571); (G) *Terribacillus aidingensis* CGMCC 1.8913 (jgi.1085072), *Terribacillus halophilus* DSM 21620 (jgi.1085769); (H) *Salimicrobium album* DSM 20748 X90834, *Salimicrobium salexigens* DSM 22782 (jgi.1096523); (I) *Halobacillus trueperi* DSM 10404 (AJ310149), *Halobacillus litoralis* SL-4 (X94558), *Halobacillus dabanensis* D-8 (AY351395); (J) *Marinococcus halotolerans* NBRC 106070 (AB682353), *Marinococcus luteus* DSM 23126 (jgi.1089306), *Marinococcus tarijensis* SR-1 (JQ413413), *Marinococcus salis* 5M (LN879357), *Marinococcus halophilus* KCTC 2843 (NPFA01000042); (K) *Salibacterium halochares* MSS4 (AM982516), *Salibacterium qingdaonense* CM1 (DQ115802), *Salibacterium lacus* GSS13 (KX818201), *Salibacterium halotolerans* S7 (LN812017), *Salibacterium nitratireducens* SMB4 (LT161881), *Salibacterium aidingense* 17-5 (DQ504377); (L) *Brachybacterium paraconglomeratum* LMG 19861 (AJ415377), *Brachybacterium conglomeratum* NCIB 9859 (X91030); (M) *Brachybacterium vulturis* VM2412 (CP023563), *Brachybacterium ginsengisoli* DCY80 (CP023564); (N) *Brachybacterium vulturis* VM2412 (CP023563), *Brachybacterium ginsengisoli* DCY80 (CP023564); (O) *Kocuria oceani* FXJ8.095 (JF346427), “*Kocuria sediminis*” FCS-11 (JF896464); (P) *Halomonas hydrothermalis* Slthf2 (AF212218), *Halomonas andesensis* LC6 (EF622233); (Q) *Halomonas alkaliantarctica* CRSS (AJ564880), *Halomonas boliviensis* LC1 (JH393258); (R) *Halomonas ventosae* Al12 (AY268080), *Halomonas mongoliensis* Z-7009 (AY962236); (S) *Halomonas pacifica* NBRC 102220 (BJUK01000094), *Halomonas salifodinae* BC7 (EF527873); (T) *Halomonas kenyensis* AIR-2 (AY962237), *Halomonas daqingensis* DQD2-30 (EF121854). The numbers in bold indicate the a posteriori probability support of the branch. The bar represents substitutions per nucleotide position. The strain names with quotation marks indicate no validated names.

**Figure 2 microorganisms-12-00482-f002:**
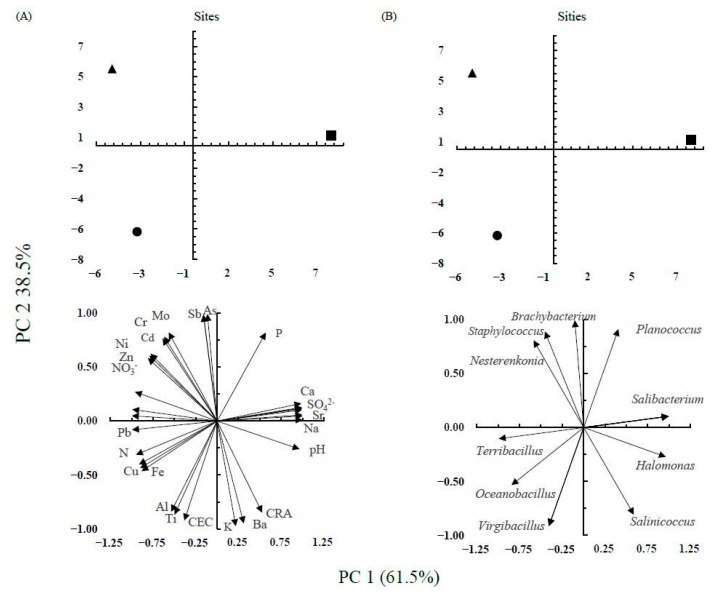
Principal component analysis (PCA) of (**A**) physicochemical characteristics of “Los Negritos” soils; (**B**) all bacterial isolated strains. Arable 1 Soil (●), Saline Soil (■), and Arable 2 Soil (▲).

**Figure 3 microorganisms-12-00482-f003:**
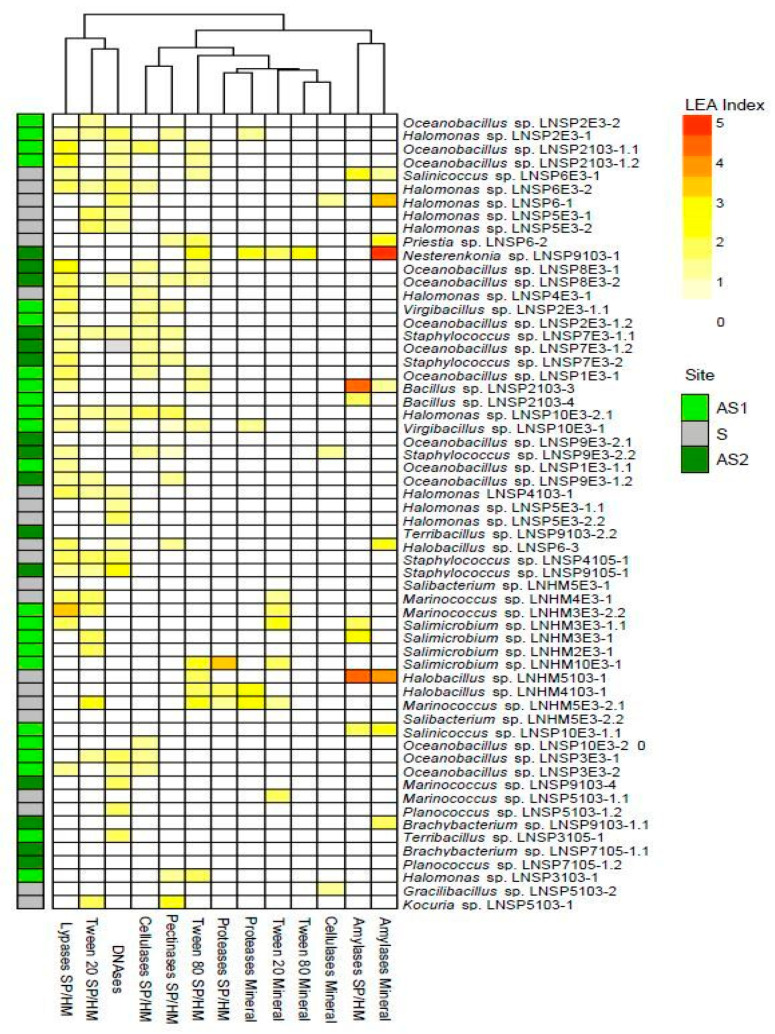
Heatmap of the levels of enzymatic activity (LEA) of the isolated strains assayed under 15% (*w*/*v*) NaCl.

**Figure 4 microorganisms-12-00482-f004:**
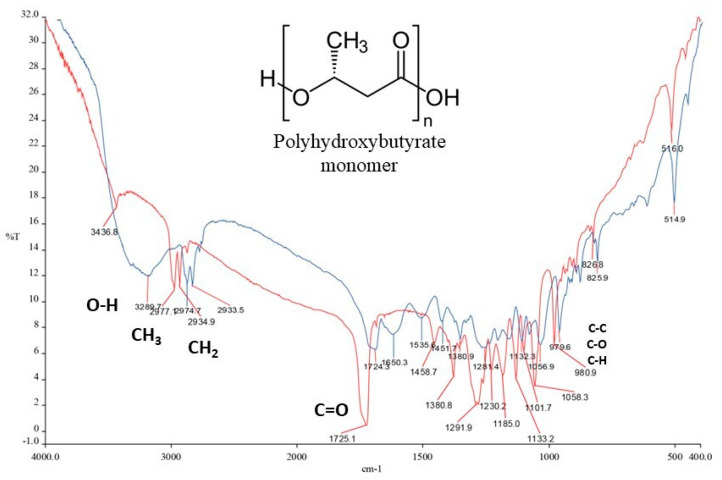
Fourier-transform infrared (FT-IR) spectra of polyhydroxybutyrate (PHB) extracted from *Halomonas* sp. LNSP3E3-1.2 (blue lines) and a pure PHB standard (red lines).

## Data Availability

The 16S rRNA sequences of the isolated strains were submitted to the GenBank database and provided with accession numbers OR553742 to OR553803.

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
