# Peer review of "Diversity and Biotechnological Potential of Cultivable Halophilic and Halotolerant Bacteria from the “Los Negritos” Geothermal Area"

_microorganisms, 2024, doi:10.3390/microorganisms12030482_

Round 1
Reviewer 1 Report
Comments and Suggestions for Authors
The authors of the paper entitled"Diversity and biotechnological potential of cultivable halo philic and halotolerant bacteria from the “Los Negritos” geo- thermal area " have done interesting work . I recommend acceptance of the paper after minor revision in the English fluency and quality of the figures if possible.
The authors of the paper entitled “Diversity and biotechnological potential of cultivable halophilic and halotolerant bacteria from the “Los Negritos” geothermal area” have done an interesting work . In their investigation the authors identified and and characterized cultivable halophilic and halotolerant bacterial diversity in a saline-sodic soil without agricultural use and two soils destined for agricultural practices as well as assessed the potential of saline and sodic soil residing microorganisms for producing biomolecules of biotechnological interest like indole acetic acid (IAA), siderophores, exopolysaccharides, PHAs, and enzymes. This is a novel work of great scientific interest, as isolating bacterial and archaea with high versatility in producing the biomolecules in high salinity environment, in the face of growing soil salinization which is predicted to affect more than 50% cultivable soil by the year 2050, is of great scientific significance. In their study they have isolated several strains that showed the ability to produce PHAs, EPS d IAA and siderophores under saline conditions (10%), which is very promising study. The strains isolated in the study they have also been reported to be capable of synthesising metabolites such as hydrolytic enzymes, EPS, and PHAs and have PGP characteristics (IAA, siderophores) under saline conditions, which are of great biotechnological interest with a wide range of applications. The paper is well written and exhaustively investigated. I have some minor comments to be addressed for the paper to be published in the journal of microorganisms.
Page 2 line 64 , ….. and to characterize the microorganisms’ potential to produce …… ? should be corrected to …… and to investigate the microorganisms’ potential to …..
Pgae 2 line 78 ……….to recover …… should be preferably changed to ……. to isolate ………..
Page 4 section 2.7. Extracellular polymeric substance production
Line number 164-165 ….A clear zone around the colonies after the addition of Lugol’s iodine stain solution was considered to indicate positivity for enzyme synthesis …..This section is describing the tests on extracellular polymeric substance production test , but the authors are referring to enzyme synthesis test on the lines I referred above, this section is a bit confusing and needs more explanation as it confuses the readership .
The reference are relevant and the conclusion is well written, however I suggest that they reduce the reference list to some extent.
Comments on the Quality of English Language
I recommend acceptance of the paper after minor revision in the English fluency and quality of the figures if possible.
Author Response
Mexico City, February 19th, 2024
To whom it may concern,
We would like to submit the revised manuscript “Diversity and biotechnological potential of cultivable halo-philic and halotolerant bacteria from the “Los Negritos” geo-thermal area” by Joseph Guevara-Luna et al. for publication in Microorganisms. We have tried to answer all comments and questions made by the reviewer and included all her/his suggestions in the revised manuscript. Please, find below the comments made by the reviewer and our answers below.
We hope that this makes our manuscript acceptable to be published in Microorganisms.
Sincerely yours
Joseph Guevara-Luna and María Soledad Vásquez-Murrieta
On behalf of all coauthors
Answers to the reviewers
We are greatly thankful for all the comments and observations that helped us to improve our manuscript. We give our reply to your comments in the following section.
- Reviewer 1
- Page 2 line 64 , ….. and to characterize the microorganisms’ potential to produce …… ? should be corrected to …… and to investigate the microorganisms’ potential to …..
A: We corrected this sentence according to the reviewer suggested as shows line 68.
- Pgae 2 line 78 ……….to recover …… should be preferably changed to ……. to isolate
A: We corrected this sentence according to the reviewer suggested as shows line 82.
- Page 4 section 2.7. Extracellular polymeric substance production
Line number 164-165 ….A clear zone around the colonies after the addition of Lugol’s iodine stain solution was considered to indicate positivity for enzyme synthesis …..This section is describing the tests on extracellular polymeric substance production test , but the authors are referring to enzyme synthesis test on the lines I referred above, this section is a bit confusing and needs more explanation as it confuses the readership .
A: We attended the reviewer’s suggestion and decided to remove the sentence above mentioned to avoid confuse the readership. Alginate lyases are enzymes that can be used as a mechanism to break down the EPS structure, but this does not fit in the meaning of the section 2.7. Now section 2.7 is showed by lines 163 – 168.

Reviewer 2 Report
Comments and Suggestions for Authors
1. Line 35. Halophiles can be divided into …… (iv) non–halophilic ….. and (v) halotolerant, which do not require NaCl for growth but can tolerate even high concentrations of this and other salts [1,3]. Non-halophilic and halotolerant microorganisms are not halophiles. Correct the sentence.
2. Line. 58. However, there are few reports on the diversity of archaea and bacteria [15] and the bacteria associated with the rhizosphere of halophytes [16]. Add few sentences to describe the results published in the works [15] and [16].
3. Line 143. Describe the method to determine IAA concentration.
4. Line 248. “analyzed physicochemical characteristics of the soils”. Include the data on soil properties in the article.
5. Figure 2. Enlarge diagrams as they are not readable.
6. In general, to improve the article:
- compare the results of your study with those obtained in similar works;
- describe the novelty of your work, as all isolates obtained are representatives of well-known microbial groups;
- describe the possibility of practical use of the strains isolated, for example, in agriculture, etc.
Comments on the Quality of English LanguageText contains some typos.
Author Response
Mexico City, February 19th, 2024
To whom it may concern,
We would like to submit the revised manuscript “Diversity and biotechnological potential of cultivable halo-philic and halotolerant bacteria from the “Los Negritos” geo-thermal area” by Joseph Guevara-Luna et al. for publication in Microorganisms. We have tried to answer all comments and questions made by the reviewer and included all her/his suggestions in the revised manuscript. Please, find below the comments made by the reviewer and our answers below.
We hope that this makes our manuscript acceptable to be published in Microorganisms.
Sincerely yours
Joseph Guevara-Luna and María Soledad Vásquez-Murrieta
On behalf of all coauthors
- Reviewer 2
- Line 35. Halophiles can be divided into …… (iv) non–halophilic ….. and (v) halotolerant, which do not require NaCl for growth but can tolerate even high concentrations of this and other salts [1,3]. Non-halophilic and halotolerant microorganisms are not halophiles. Correct the sentence.
A: We corrected the sentence in lines 35 - 38 as fallow:
Halophiles can be divided into (i) extreme (2.5–5.9 M ∼15–32% NaCl), (ii) moderate (0.5–2.5 M ∼3–15% NaCl), and (iii) slight ( 0.2–0.5 M ∼1–3% NaCl), however, there are halotolerant microorganisms, which do not require NaCl for growth but can tolerate even high concentrations of this and other salts [1,3].
2. Line. 58. However, there are few reports on the diversity of archaea and bacteria [15] and the bacteria associated with the rhizosphere of halophytes [16]. Add few sentences to describe the results published in the works [15] and [16].
A: We attend the suggestions of the reviewer and added a short and pertinent description of the results in the works [15] and [16] that can be find in lines 58 - 64 as follow:
“Los Negritos” is a geothermal area that has been studied in terms of soil and water characterization to evaluate salt content, as well as geothermal activity as a potential alternative energy source [13,14]. There are few reports on the diversity of microorganisms inhabiting the soils of the geothermal area. Guevara-Luna et al. [15] investigated the archaea and bacteria community diversities throughout high-throughput sequencing analysis of the 16S rRNA and the physicochemical characteristics that could explain their relative abundance in these soils. On the other hand, Pérez-Inocencio et al. [16] isolated the bacteria associated with the rhizosphere of halophytes in this area, which showed different PGP activities and synthesis of hydrolytic enzymes.
3. Line 143. Describe the method to determine IAA concentration.
A: We describe the method in lines 154 - 157 as fallow:
The cell-free supernatants were obtained at 4,500 rpm and an aliquot of 1 mL was mixed with 1 mL of the Salkowski solution. The IAA standard (Sigma-Aldrich) was used to construct a calibration curve from 0 to 1000 µg mL-1. Azospirillum brasilense Sp7 was used as positive control without NaCl addition.
- Line 248. “analyzed physicochemical characteristics of the soils”. Include the data on soil properties in the article.
A: We cannot add the data on soil properties in the article because these are published in other article and protected by copyright, but we cited the reference where the data can be consulted.
- Figure 2. Enlarge diagrams as they are not readable.
A: We enlarge and improve the diagram for a better reading.
- In general, to improve the article:
- compare the results of your study with those obtained in similar works;
A: We tried to include and compare the results from similar works and exclude the information no relevant to discuss the information of this article.
- describe the novelty of your work, as all isolates obtained are representatives of well-known microbial groups;
A: We describe the importance of the strains isolated in this works besides there are grouped in well-known microbial groups in the lines 387 to 390.
Furthermore, the strains isolated in this work are representatives of well-known microbial groups, however, the findings in this study suggest that these could be new representatives of these microbial groups and more detailed studies are required to confirm that observation.
- describe the possibility of practical use of the strains isolated, for example, in agriculture, etc.
A: We described the practical use of the strains isolated as indicate in lines 390 to 395:
On the other hand, the findings in this study indicate that the strains or their metabolites could be applied to agriculture in soils with high salt concentrations to improve the development of crops; The hydrolytic enzymes are promising for industrial applications such as development of biodegradable detergents or treatment of residual water, but more studies are needed.

Round 2
Reviewer 2 Report
Comments and Suggestions for Authors
Authors described most of questions and improved the article.
Few additional commentaries:
1. Line 154: “The cell-free supernatants were obtained at 4,500 rpm and an aliquot of 1 mL was mixed with 1 mL of the Salkowski solution. The IAA standard (Sigma-Aldrich) was used to construct a calibration curve from 0 to 1000 µg mL-1. Azospirillum brasilense Sp7 was used as positive control without NaCl addition.”
Add the method used (spectrophotometry, chromatography, etc.?) and the equipment.
2. Line 250. We cannot add the data on soil properties in the article because these are published in other article and protected by copyright, but we cited the reference where the data can be consulted.
Indeed, you cannot double the data published in other work but these data should be cited and described briefly in your work. For example, “In the work [15], it was shown that the following soils were find in the area….”
Author Response
Answers to the reviewers
We are greatly thankful for all the comments and observations that helped us to improve our manuscript. We give our reply to your comments in the following section.
- Reviewer 2
- Line 154: “The cell-free supernatants were obtained at 4,500 rpm and an aliquot of 1 mL was mixed with 1 mL of the Salkowski solution. The IAA standard (Sigma-Aldrich) was used to construct a calibration curve from 0 to 1000 µg mL-1. Azospirillum brasilense Sp7 was used as positive control without NaCl addition”.
Add the method used (spectrophotometry, chromatography, etc.?) and the equipment.
A: We added the recommendations made by the reviewer
The cell-free supernatants were obtained at 4,500 rpm and an aliquot of 1 mL was mixed with 1 mL of the Salkowski solution [37]. The IAA standard (Sigma-Aldrich) was used to construct a calibration curve from 0 to 1000 µg mL-1 spectrophotometry at 540 nm using a Hach DR 2800. Azospirillum brasilense Sp7 was used as positive control without NaCl addition.
- Line 250. We cannot add the data on soil properties in the article because these are published in other article and protected by copyright, but we cited the reference where the data can be consulted.
Indeed, you cannot double the data published in other work but these data should be cited and described briefly in your work. For example, “In the work [15], it was shown that the following soils were find in the area….”
A: We considered the observations and recommendations made by the reviewer and we rewrite the paragraph for a better understanding and readership that can be find in lines 251 – 262 as follow:
The “Los Negritos” soils showed some interesting characteristics such as cation exchange capacity (CEC), total nitrogen (N) content of AS1 soil showed a value of 46.8 cmol kg-1 and 1.8 g kg-1 dry soil (respectively). The saline soil showed a pH value of 9.2 and electrolityc conductivity (EC; 17.6 dS m-1), while the AS2 soil showed high content of arsenic (As; 175 mg kg-1 dry soil), cadmium (Cd; 5.7 mg kg-1 dry soil) and nitrates (NO3-; 39.7 mg kg-1 dry soil) [11]. Considering the above values, in this work the PCA analysis showed that the analyzed physicochemical characteristics of the “Los Negritos” soil explained 100% of the data variation and the association between these and the isolated strains (PC1 61.5% and PC2 38.5%; Figure 2). Members of Oceanobacillus, Terribacillus, and Virgibacillus were enriched in the AS1 soil, while members of Halomonas Salibacterium, and Salimicrobium were enriched in the saline soil, and members of Brachybacterium, Nesterenkonia and Staphylococcus were enriched in AS2 soil (Figure 2).
